# Characterizing 'New Korean Confucianism': Focusing on Pak Chonghong and Yi Sang-ŭn's Life and Thought

**Xing-Ai Gao [1,\*] and So-Yi Chung [2,\*]**

1 Department of Ideological and Political Education, School of Marxism, Yanbian University, Yanji 133002, China

2 Department of Religious Studies, School of Humanities, Sogang University, Seoul 04107, Republic of Korea

\* Correspondence: goosungae@ybu.edu.cn (X.-A.G.); soyichung@sogang.ac.kr (S.-Y.C.)

**Abstract:** This article attempts to characterize some important aspects of 'New Korean Confucianism,' by focusing on the life and thoughts of two major thinkers of 20th century Korea, namely Pak Chonghong (1903–1976) and Yi Sang-ŭn (1905–1976). While there are volumes of studies on 'New Confucianism', the focus remains mostly on Chinese academics; however, just like Song–Ming, Neo-Confucianism spread throughout East Asian countries, where a unique and distinct Neo-Confucian model emerged with its own arguments and debates. The New Confucianism that appeared at the turn of the 20th century in an attempt to embrace Western cultural power within a Confucian value system has also been extended and widely adopted in Korea and has transformed itself according to its socio-political environment. Pak and Yi, who lived during and after the Japanese colonial period, struggled to grasp a sense of autonomy as the unique Korean tradition and spirit was about to be swept away by a flood of foreign ideas and social turmoil. Although the two thinkers differed in their approaches—Pak studied Western philosophy and Yi Chinese New Confucianism—they devoted their life to uncovering and systematizing the distinctive structure of Korean traditional philosophy and thereby laid the cornerstone of New Korean Confucianism.

**Keywords:** Korean Confucianism; New Confucianism; Pak Chonghong; Yi Saung-ŭn; 20th century Korean philosophy; traditional Korean Thoughts; Korean autonomy

## 1. Introduction

The notion of 'New Confucianism' emerged out of the need to completely renew one's traditional identity after China's defeat in the Sino-Japanese War of 1894. Japan had opened itself up to Western technology and philosophy much earlier than Korea and China, and its victory over China and the subsequent annexation of Korea in 1910 clearly showed the power of the 'West' (Japan) over the 'East' (China and Korea).

On the threshold of the 20th century, China's cultural pride was deeply hurt; not only by Japan, but even more so by the European invaders who forced China into a semi-colonial position. It caused Chinese intellectuals to take up the brushes to denounce the long-held Confucian tradition as outdated and urged for the full acceptance of Western civilization. Meanwhile, a number of Confucian thinkers firmly held that it was not the Chinese Confucian value system per se that was the culprit. All they needed was an internalization or absorption of the Western cultural system *within* Chinese traditional values. Thinking back, Chinese Confucianism had already been through such a transformation twice: first, Classical Confucianism during the Han dynasty unified various streams of thought such as Legalism and Mohism; second, Neo-Confucianism during the Song–Ming dynasties embraced the Buddhist framework and created its own ethical universe. Despite the incessant flux of alien thoughts, Confucianism managed to keep its identity intact while meeting the demands of the times. Now, all that was needed was the courage to learn from the West and absorb their essential teachings without losing one's identity and self-esteem.

For this, Chinese intellectuals of the time launched the third Confucian transformation named, 'New Confucianism.'[1] Although there may be a range of ways to define such a trend, 'New Confucianism' was born with three fundamental features: first, affirmation of Confucianism as the core spirit of Chinese culture amongst battling ideas; second, an active understanding of modern Western culture in order to revitalize Confucian tradition, both in its format and its content; third, through such adaptation and consilience, its purpose was to give a reasonable perspective and viable solutions to problems of the time, viz., recover their wounded pride and restore self-esteem.[2]

Korean intellectuals underwent the same humiliation and crash of self-respect under Japanese imperial rule. Long-held Confucian values were stripped of their authority, while the full-scale acceptance of Western cultural products was forced upon Koreans throughout the Japanese colonial period. Due to the circumstances of the time, the study of philosophy under Japanese occupation was often devalued as state-led studies or as collusions with the Japanese imperial government. In spite of these historical restrictions, however, a considerable number of philosophers actively pursued ways to confront the problems and paradoxes of the colonial reality. In the 1920s, a wave of philosophers who studied abroad in Japan, China, and the United States returned to Korea and were later joined by graduates of Keijō Imperial University in the 1930s. Together, they constituted the first generation of Western philosophy scholars in Korea,[3] and while devoting themselves to the study of Western theories, they also continued to scrutinize the relationship between philosophy and Korea's colonial reality, and agonized over the role and identity of philosophy under Japanese occupation. Despite Japan's censorship that prevented free and autonomous research, these scholars conducted themselves with a clear awareness of the problems and tasks that arose from Korea's situation, thus laying the foundation for modern philosophy in Korea. On the other hand, the first-generation scholars reinforced the dichotomy between Western philosophy and traditional thought by making an either–or choice, and their acceptance of Western ideas left no room for traditional philosophy in their scholarship. In such circumstances, there were still exceptions: some first-generation scholars pursued their research in traditional ideology and pondered ways to preserve and reinterpret Eastern philosophy. They were Pak Chonghong (朴鍾鴻; penname Yŏl-am 洌巖, 1903–1976) and Yi Sang-ŭn (李相殷; penname Kyŏngro 卿輅, 1905–1976), two seminal thinkers of 20th century Korean philosophy who also left distinct legacies in modern Korean history.[4]

This article aims to compare the academic trajectories of Pak Chonghong and Yi Sang-ŭn[5] and to identify them as the pioneering scholars of 'New Korean Confucianism'.[6] Early in their careers, the two contemporaries followed vastly different paths. Pak, for instance, studied philosophy at Keijō Imperial University in Korea, while Yi studied abroad at Peking University where he was exposed to various Western theories and the complexities of Chinese academia such as New Confucianism. After Korea's liberation, Pak concentrated on studying Western philosophy at Seoul National University, whereas Yi focused his research on Chinese philosophy at Korea University. However, both eventually shifted to the study of Korean thoughts and philosophy in their research. The article will go on to expound the issues these two philosophers raised about colonial reality, the efforts they made to rebuild academia after Korea's liberation, and their motivations in choosing to refocus their research on Korean philosophy. A comparative study of Pak and Yi will reveal the distinct features of their philosophies and situate them within the larger context of New Confucianism in contemporary East Asia.

## 2. Reality and Philosophy under Colonial Rule

History bestows each era with its own task. The March 1st Movement—the first major mass-movement for Korean independence—imbued Pak Chonghong with a sense of national consciousness. The harsh reality of colonial rule motivated him to pine after an understanding of Korea as a shared and collective 'nation' (民族).[7] His first article published in 1922, "*Chosŏn misulsa mijŏnggo* (朝鮮美術史未定稿 *An Unfinished Manuscript on Chosŏn's*

*Art History*)," reflects his attempt to uncover a shared trait that distinguishes the people of Korea from other ethnicities. However, his limited knowledge of aesthetics thwarted additional publications on art history, and his interest in aesthetics evolved into an interrogation of more fundamental issues (Gao 2013, pp. 31–55).

In April 1929, Pak entered the Department of Philosophy at Keijō Imperial University[8] as an undergraduate and studied Western philosophy under Japanese professors who had studied abroad in Europe. They were open-minded and liberal scholars, influenced by philosophical trends such as Heidegger's existentialism and New Hegelianism.

Under the tutelage of Professor Miyamoto Wakichi (宮本和吉, 1883–1972)[9], Pak completed his senior thesis, *On the Notion of Sorge in Heidegger* (ハイデッガーに於けるSorge について) (Pak 1998, vol. 1, pp. 161–249), and graduated from the College of Law and Humanities in March 1933. He was one of the fifth batch of graduates from Keijō Imperial University, along with colleagues such as Ko Hyŏnggon (高亨坤, 1906–2004, former professor at Seoul National University), Yi Kapsŏp (李甲燮, defected to North Korea), and Pak Ch'i-u (朴致祐, 1909–1949) (Kim 2007, pp. 179–217).

In addition to his senior thesis, Pak's research on Heidegger includes his unpublished article "The Internal Possibility of Transcendence in Heidegger" (ハイデッガーに於ける 超越の内面的可能性について), and "The Issue of Horizons in Heidegger" (ハイデッガー に於ける地平の問題), published by the periodical *Yisang* (理想) in 1935. Through his research on Heidegger, Pak opened his eyes to the reality of his nation and discovered the true nature of the power that subjugated Korea, and this led him to investigate ways to confront the problems that arose in his nation.

Pak borrows Heidegger's notion of *Geschehen* (occur) to explain what it means to 'philosophize': he argues that philosophizing should begin with "issues that are the most direct and concrete to us" and that it should be grounded in "this era, this society, this land, and this reality of existence itself" (Pak 1998, vol. 2, pp. 322–31). According to Pak, philosophy is that which inevitably surges out of the anguish one feels towards the unjust and desperate reality under colonization, and to philosophize is to rigorously investigate the problems of reality and seek solutions.

The crisis of colonization presented no other options for Pak. The problems that arose from reality were not merely intellectual puzzles but the most fundamental and desperate tasks of philosophy. The crisis of reality required a philosophy that was critical of reality, and Pak argued that 'our philosophy' was inseparable from reality. At the same time, Pak discovers Heidegger's basic existentialism to be one-dimensional and abstract. Heidegger's theory lacked the call for specific action and was limited to individual self-awareness, and Pak hence suspended his research on the German philosopher.[10]

Pak was not the only philosopher who agonized over his nation's loss of sovereignty. This was a task bestowed by history and nationalism that motivated many of Pak's contemporaries, especially after the March 1st Movement. (Chŏng 1959, pp. 832–3). Unlike Pak who had no financial means to study overseas, Yi Sang-ŭn began his academic training in China at the age of 18. Yi was encouraged by his father and grandfather's strong belief in the slogan 'revere Confucianism and renounce Japan' (崇儒排日). From 1925 to 1931, Yi enrolled in preparatory and regular courses at Peking University and graduated with a degree in philosophy.

Back then, Peking University was an institution of the utmost prestige, and it evolved into a liberal community that served as the hub of the New Culture Movement (新文化運動). During the 1920s, multiple scholars such as Hu Shi (胡適, 1891–1962) who had studied at Western institutions were employed at Peking University, and they spurred a lively academic climate with their research on logic, pragmatism, Lebensphilosophie, and New Realism. At the same time, the clash of Eastern and Western theory led to a critical reflection on traditional culture, and many Chinese philosophers began to question the role and path of Chinese philosophy (Hong 2006).

Yi recalls how one of the courses he took during his preparatory training, Survey of National Heritage (國故概要), played a pivotal role in his decision to major in philosophy

and his plan for future research projects. (Yi 2018, vol. 1, pp. 225–31) In fact, Yi displayed keen interest in the movement to 'systematize national heritage' (國故整理), a project led by Hu Shi to restructure the classic canons of the East based on philological and empirical methodology (Kim 2006; Lee 2011). He was an avid reader of Hu Shi's *Outline of the History of Chinese Philosophy* (中國哲學史大綱, 1919) and *The Writings of Hu Shi* (胡適文存, 1921), and also perused Liang Qi-chao's (梁啓超, 1873–1929) *An Outline of Scholarship in the Qing Period* (清代學術概論, 1920), *A History of Pre-Qin Political Thought* (先秦政治思想史, 1922), and *A History of Chinese Academic Thought in the Past Three Hundred Years* (中國近三百年學術思想史, 1924) (Yi 2018, pp. 3–35).

Yi also studied *Eastern and Western Cultures and their Philosophies* (東西文化及其哲學) by Liang Shuming (1893–1988), *Debates on Ancient History* (古史辨) by Gu Jiegang (顧詰剛, 1893–1980), *A Chronological Study of Pre-Qin Philosophers* (先秦諸子繫年) by Qian Mu (1895–1990), and *A History of Chinese Philosophy* (中國哲學史) by Feng Youlan (1895–1990). Through these monographs, Yi discovered the need to reinterpret and reevaluate traditional Eastern thought; soon, he became drawn to the methods of New Confucianism that pursued a modern application of Confucianism by absorbing and synthesizing Western culture.

When he finished his studies in China and returned to Korea, Yi was determined to apply his education and initiate the campaign to 'systematize Korea's national heritage', namely to "uncover the Eastern philosophy that Korea embraced and cultivated into its own thought and the Eastern philosophy that is inherent to Korea, to systematize and theorize it from a modern perspective so as to properly reevaluate and rediscover ourselves" (Yi 2018, p. 229). However, the conditions of colonization had completely removed the authority of traditional Eastern philosophy, and Yi had to settle for lesser tasks such as serving as the Secretary of Korean Language (韓文秘書) at the Consulate General of China, and teaching Chinese and introductory courses on Kant at Posŏng College (普成專門學校).

## 3. The Flourishing of Philosophical Research within the Liberated Space

After Korea's liberation in 1945, Pak Chonghong secured a job at Seoul National University where he concentrated on his teachings and writings. He also set out to establish a system of Western philosophy with a special focus on logic. Pak's posthumous work reveals his attempt to design a system that begins with general logic and epistemic logic, moves on to dialectics and the logic of inversion, and culminates in the logic of creation.[11] This system was based on Pak's emphasis on logic as the foundational theory that upholds the methodology of every academic field.

Pak especially focused on Hegel's dialectical logic that consists of the logic of being (die Logik des Seins), the logic of essence (die Logik des Wesens), and the logic of notion (die Logik des Begriffs). In Pak's reading of Hegel, essence is considered to be the negation of being, and notion is considered to be the negation of the negation of being. The immediacy of being and the reflection of essence gives rise to notion, and notion 'preserves' (保維 erhalten), in an elevated state, the being and essence that has 'gone under' (沒落 untergehen). Therefore, notion is 'the product of being and essence,' the synthesis of being and essence, and the truth of being and essence.[12]

If we were to take being as a thesis, then essence—the negation of being—would be the antithesis, and notion would be the synthesis of being and essence. Pak writes that "the dialectics of being deals with the transformation of being, the dialectics of essence deals with the system of reflections, and the dialectics of notion deals with the development (Entwicklung) that arises from how this conflict returns to the transformation of being." (Pak 1998, vol. 3, p. 520) This led him to focus on the dialectics of notion as a system of synthesis.

Pak applies this framework of dialectics to his interpretation of modern Western philosophy. Taking his cue from Carl Jung's categorization of the human psyche into introversion and extroversion, Pak categorized the trend of modern Western philosophy into introspective existentialism and the extroverted philosophy of science.[13] According to Pak, existentialism concentrates on emotional aspects more than logic, and attempts to under-



stand reality through the autonomous inner life of humans instead of the external object world. On the other hand, the philosophy of science refuses to rely on the trial and errors of empiricism, and instead concentrates on rationality and meticulous scientific methods (Pak 1998, vol. 2, pp. 317–28, 455–80).

If we were to take introspective existentialism as a thesis, then the extroverted philosophy of science would be the antithesis. However, according to Pak, the two are "not counterpoints, but one-dimensional abstractions of what is inherently a single path." (Pak 1998, vol. 2, p. 478) In other words, they are fundamentally the two sides of a single path. Western philosophy's failure to recognize this is what led the field to reach its impasse—the philosophy of science fixates solely on science, and existentialism is trapped in the world of abstraction. Additionally, it is Eastern philosophy that, as the synthesis, has the potential to embrace both aspects as one.

While Pak Chonghong was leading the study of Western philosophy in Korea's post-liberation academia, Yi Sang-ŭn paved the way for research on Chinese philosophy. While teaching and researching Chinese philosophy at Korea University, Yi set out to reinterpret Confucianism from a modern perspective and discover its modern significance. Elements of this project include Yi's various research papers on Chinese philosophy[14] and an unfinished manuscript on the history of Chinese philosophy written in the form of lecture notes.[15]

Ever since the movement to 'systematize national heritage,' Chinese classics have taken on a new life through modern reinterpretations, and Yi argues that the study of classics must be based on scientific reasoning and sufficient bibliographic evidence.[16] For instance, in Yi's analysis of Mengzi's theory of the innate goodness of human nature, he cites the classic Shijing (詩經) and Kong Ying-da's (孔穎達) annotation to demonstrate that the difference in Mengzi and Gaozi's stance on human nature derives from their different understandings of the notion of life (生) (Yi 1998, vol. 3, p. 338)

However, objective and logical analyses based on 'scientific' methods[17] alone are not enough—they must be accompanied by sympathetic and empirical methods,[18] and researchers should engage in rational interpretations by investigating the innate and essential value of life in the classics. Yi argues that researchers must "return to the 'human spirit' that transcends time, and adopt an empathetic and empirical method when reading the later scholars' in order to grasp the true spirit of our predecessors." (Yi 1998, vol. 4, p. 187) This, according to Yi, was the way to revive spiritual energy in modern times.[19]

After presenting a rational mode of interpretation that integrates the objective and logical methods of science and the subjective mode of empathy and empiricism,[20] Yi began to apply this to the systematization of the history of Chinese philosophy. According to Yi, Chinese philosophy began with Confucius, the one who "compiled all the ideology that came before the Eastern Zhou dynasty." Instead of heralding Laozi, a philosopher that barely left a trace in history, Yi lionizes Confucius as the one who established the tradition of pursuing academic studies (講學) and accomplished the great cause of "inheriting the scholarship of past sages and building the path for future scholarship." (Yi 1998, vol. 3, pp. 10–11).

Yi goes on to categorize the history of Chinese philosophy into ancient philosophy, medieval philosophy, and early modern philosophy; and further subdivides the history into the Period of Masters (*Zixue* 子學) and the Period of Classical Studies (*Jingxue* 經學).[21] Ancient philosophy encompasses the 400-year span from the late Spring and Autumn period to the early Han (漢) dynasty, and because it includes the Hundred Schools of Thought (諸子百家), Yi defines it as a kaleidoscopic period that built the foundation for future philosophical theories. The following 700-year span from the early Han dynasty to the Sui (隋) and Tang (唐) dynasty is the period of medieval philosophy, witnessing the rise of Classical Studies and Dark Learning (*Xuanxue* 玄學) in Confucianism along with Buddhism, which led to a fusing of traditional and foreign thought. The 1000 years that ranges from the late Tang/early Song (宋) to the late Qing (淸) dynasty is the period of early modern philosophy, a time when Neo-Confucianism arose as a reaction against Buddhism and Daoism,

while Shixue (實學) and the school of Kaozheng (考證學) emerged as a reaction against Neo-Confucianism.[22]

With Confucianism as the common thread, Yi traces the changes and evolution of the history of Chinese philosophy. Instead of looking for gaps or ways to surmount the history, Yi focuses on inheriting and expanding the tradition. Despite being unfinished[23]—Yi only completed the section on ancient philosophy, namely the part on the emergence of the Hundred Schools of Thought and the school of Confucianism that ranges from Confucius to Mengzi and Xunzi—the project seeks to move beyond a mere chronology of schools of philosophy (學案) and towards a history of philosophy that is based on logic, objectivity, and rational interpretation.

### 4. The Transition to Research on Korean Thought

The transition of Pak and Yi 's academic interests to Korean thought in the late 1950s was the result of a complex interplay between various factors. One factor was the prevalent outlook on Korea's reality among contemporary intellectuals. Despite Korea's independence, the country was still under a poverty-ridden, unfair reality and faced the danger of either being colonized by yet another foreign power or succumbing to North Korea and communism. This fear and anxiety were often abbreviated into an awareness of Korea's 'backwardness,' which instigated the obsession over catching up with the West (Lee 2014, vol. 116, p. 450).

Korean intellectuals were engrossed in learning and imitating Western philosophy, and traditional philosophy was considered to be the cause of Korea's downfall and a relic of the past. It was not until the mid-1950s that scholars could afford to look back and reflect upon traditional thought. Cultural nationalists led the discourse on how to restore and reorganize the study of traditional thought that had been discontinued under Japanese colonialism.

At this point, Pak and Yi were simultaneously presented with the opportunity to witness and experience the West for themselves: invited by the U.S. Department of State, they each spent a year in the U.S. as visiting scholars.[24] During this time, they were able to witness trends in the study of modern Western philosophy and realized that modern Western civilization was not without shortcomings; further, they glimpsed that Eastern philosophy could, albeit unstructured in its present form, give some viable solutions to the social ills of modern global society. In September 1958, Pak and Yi attended the 12th World Congress of Philosophy held in Venice, Italy, where they bore witness to the rising role of Eastern philosophy in the advancement of world philosophy.

Consequentially, Pak and Yi recognized that the study of Eastern thought—Korean thought in particular—was an urgent task and changed their course of research. July 1957 saw the publication of the inaugural issue of *Han'guk sasang* (Korean Thought), an academic journal of a newly formed society which aimed at cultivating independent research on Korean thought. Pak and Yi joined the journal as editorial advisors from its second issue onwards. Han'guk Sasang Yŏn'guhoe (The Society for Research on Korean Thought) was founded in February 1963, with Pak as its first president and Yi as an editor. The early activities of the society focused on raising Korea's national pride. They identified aspects of traditional thought that corresponded to Western philosophy, and promoted the idea that the works of Korean philosophers were on par with that of Plato or Kant.[25]

Through his research on Korean philosophy, Pak worked towards establishing a framework for modern Korean thought and mainly focused on the study of Korean Buddhism and Confucianism. He began to actively publish his research in the 1960s. Pak argued that Korean thought already carried the depth of insights found in the greatest philosophies of the West: it was overlooked due to differences in how the two cultures phrase and contextualize their ideas. Moreover, he strove to elevate Korea's sense of cultural pride in facing Western culture's dominance, and to create a synthesis of Eastern and Western philosophy.

Meanwhile, Yi's research was motivated by his dream of initiating Korea's own systematization of national heritage. Yi insisted that traditional thought had not yet lost its vi-

tality, and that systematizing and theorizing traditional thought through a modern and rational perspective could play a part in solving problems in the modern Westernized society. He believed that Korean philosophy would significantly contribute to what he called 'the third path of civilization' that promoted coexistence and prosperity in the world. Yi also planned to draft *Yijo yugyo ch'ŏrhaksa* (*The History of Confucian Philosophy under Chosŏn's Yi Dynasty*) to rectify Korean academia's misunderstandings of Neo-Confucianism and Silhak (實學, Practical Learning), and at the same time, compile a systematic history of Korean thought (Yi 1998).

### 4.1. Pak Chonghong's Research on Korean Thought

In fact, from the beginning of his career, Pak's main interest was Korean traditional thought. He was particularly interested in the philosophy of Yi Hwang (李滉, penname Toegye 退溪, 1501–1570), but judging that it would be difficult to study Korean philosophy under a Japanese professor at Keijō Imperial University, Pak decided to specialize in Heidegger and Western philosophy. However, after realizing the limitations of existentialism and the philosophy of science, he also witnessed Western philosophers directing their interest towards Eastern philosophy, and decided to direct his own research back to traditional thought.

Pak was highly critical of contemporary academics that flocked to new trends in Western philosophy while completely dismissing their own traditional thought. "We must be open-minded and brave in our learning and accepting of foreign ideas that are more advanced. However, in order to recognize that others are more advanced than us, should we not have to first understand ourselves?" he asks (Pak 1998, vol. 4, p. 198), and delved into traditional philosophy as a way to better understand 'ourselves.'

His research on traditional philosophy is divided broadly into Buddhist thought and Confucian thought. In studying Buddhism, Pak selected five thinkers that are both universal and creative—Sŭngrang (僧郎, c. 450–530), Wŏnch'ŭk (圓測, 613–696), Wŏnhyo (元曉, 617–686), Chinul (知訥, 1158–1210), and Ŭich'ŏn (義天, 1055–1101)—and identified values and issues commonly shared with Western philosophy to demonstrate Korea's "aptitude and competence in philosophical meditation" (Pak 1998, vol. 4, p. 198). By approaching Buddhism from the perspective of Western philosophical issues, Pak challenges the dominance of the West and reveals Korean Buddhist thought to be a worthy peer of Western ideas.[26] In other words, Pak's research on Buddhist thought "exceeds mere academic interest–it is based on passion and a pioneering spirit" (Keel 1998, p. 254).

This attempt to restore national pride continued in his research on Confucianism. Despite originating in China, Confucianism evolved into a unique system of thought in Korea. Pak especially highlights Yi Hwang who refused to blindly adhere to and obey (墨守遵奉) the Neo-Confucianism of Zhu Xi (朱子學), and instead developed his own theory based on his own philosophical speculations. Pak points to the publication of Yi Hwang's *An Outline of the Works of Zhu Xi* (朱子書節要) in Japan and the printing of *Ten Diagrams to Become a Sage* (聖學十圖) in Beijing to illustrate the international attention towards Yi Hwang's distinct philosophy (Pak 1998, vol. 4, p. 380).

Pak also maintains that the emphasis on human dignity, existentialism, and the philosophy of science are not exclusive to Western philosophy, but are features that can be found in Korean Confucianism as well. For instance, Tonghak's (東學) doctrines on human dignity advocate that man and the heaven are one (人乃天),[27] and that all individuals must serve each other the way they serve the heavens (事人如天). This lead Pak to examine the notion of 'sincere-reality' (誠實), a term that shares part of its etymology with existentialism (實存主義). According to Pak, both philosophies revolve around the notion of humans as beings that strive to be true and real (實), and that their difference is merely in the textual settings and phrasings (Pak 1998, vol. 5, pp. 6–14). Furthermore, Pak argues that while the Confucian scholar Ch'oe Han'gi (崔漢綺) "did not explicitly use the word 'philosophy,' he presented the most ground-breaking thoughts on the philosophy of science" (Pak 1998,

vol. 5, pp. 251–52) and therefore fundamentally was in line with seemingly more logical and empirical western philosophical thoughts.

Moreover, Pak highlights how Confucian thought encompasses both ethical (道學的) and practical (實學的) elements and presents this as the basis of Eastern philosophy's ascension to an alternative to contemporary Western philosophy. He regards 'sincere-reality' (誠實) and 'real-use (practicality)' (實用) to be the two sides of 'reality' (實), and points to the creative spirit of his Korean predecessors[28] who strove to create a dialectical synthesis of these two notions, the thesis being the ethical or introspective aspect of sincere-reality, and the antithesis being the economical or extroverted aspect of advancing social welfare by supplying the public with practical technology (利用厚生).

Above all, Pak appeals for the need to "revive, receive and carry on" this creative spirit in order to formulate a new, potent response to Korea's modern state of affairs. He refers to this as 'inheriting' (傳承) in his writings (Pak 1998, vol. 5, p. 534). He writes, "only when we inherit the philosophical legacy of every era based on the viewpoint of this very era, this very society, this land, and our position at our current stage and makes it as part of our own, then can we build a philosophy of our own," (Pak 1998, vol. 1, p. 385) as he stresses the importance of carrying over traditional thought.

However, this must be supplemented with imported philosophy since the consumption of foreign ideas has become an inevitable reality. If only the late Chosŏn attempts at accepting foreign ideas were properly inherited, Pak laments, Korea would have avoided the regression it experienced during the colonial period. Above all, Pak states that the consumption of foreign ideas is necessary to prevent regression and backwardness, and that scholars must be aware of what and how to consume.

Pak sheds harsh light on the chaos that followed Korea's liberation—a chaos caused by the deluge of foreign ideas that were imported without any mediation, scrutiny, or process of selection. The dregs of foreign philosophy must be filtered out, he argues, and Korea's academia should be selective in what to accept and absorb as part of their own. Pak warns against hasty imitation and blind admiration of the West, and instead promotes a bold and active approach in contributing to a global philosophy by reviving the aspects of traditional thought that jibes with the fundamental spirit of Western philosophy (Pak 1998, vol. 5, pp. 534–36).

Moreover, Pak argues that it is of utmost importance to establish a sense of national autonomy.[29] National autonomy is a product of history that is built upon the democratic values of freedom and equality. It is forged through awakenings and active engagements, and is revived over and over for it is "a spirit and power that is alive and in motion" (Pak 1998, vol. 6, p. 165). Therefore, only when foreign ideas are consumed by way of a solid sense of national autonomy can they be properly integrated into and rekindled as Korean thought, and become an idea that Korea can proudly present to the world.

Pak also displayed strong interest in the dialectical synthesis between the inheritance of traditional thought and the consumption of foreign ideas. According to Pak, it is impossible to properly inherit tradition in an environment that refuses to absorb foreign ideas, just as it is impossible to properly ingest and absorb foreign ideas in an environment that has failed to inherit their own traditional philosophy. These two movements ultimately share the objective of creating a new philosophy that sets the foundation for contemporary Korean ideas.

> Our path to survival, the road we must pioneer with our own power is the road of creation that starts with introspective awakening, collides with nothingness, and returns to extroversion. [ . . . ] The boundless forces of reality only exist in the sublation of introspection and extroversion, conscience and technology into a single, absolute form of action. This is the site that enables creation and achieves construction. (Pak 1998, vol. 2, p. 368)

In other words, scholars should not pit the introspective inheriting of traditional thought against the extroverted consumption of foreign ideas, but sublate the two in order

to create a new form of contemporary Korean thought. Or, in Hegelian terms, if introspective traditional thought is a thesis, extroverted foreign ideas would be the antithesis, and contemporary Korean thought would be the synthesis of the two. For Pak, this synthesis is the ultimate task of philosophy that must be accomplished by Koreans themselves.

*4.2. Yi Sang-ŭn's Research on Korean Thought*

From the very beginning of his career, Yi was determined to initiate Korea's own 'systematization of national heritage,' and reevaluate both the Eastern thought that Korea embraced and expanded, and the traditional thought that was unique to Korea. However, when Korea's intellectual climate reduced traditional thought to a relic of past failures, Yi had to turn his attention to Chinese philosophy. He later returned to the study of Korean thought when Korea's academia decided to scrutinize and correct their previous dismissal of Korean philosophy.

Despite the change in his object of research, Yi's original approach and spirit prevailed in his study of Korean philosophy. Yi set out to write a logical and objective account of the history of Korean thought. He reinterpreted old classics through rational methods, and explored the intrinsic logic of Korean thought from a perspective that focused on inheriting and complementing traditional philosophy. This was his project to 'systematize Korea's national heritage'—his lifelong dream and calling.

First, let us begin with an overview of Yi's understanding of Korean thought. Yi recognizes Korea to be an active agent for the advancement of philosophy, but refuses to fixate only on traditional thought. According to Yi, the dissemination of culture is vertical and top-down, but the exchange of cultures happens between nations of the same, lateral status. It is a universal fact that cultures advance through the consumption and absorption of foreign ideas, and, therefore, no matter how foreign the culture is, once it is absorbed and creatively expanded, it becomes part of the nation's own (Yi 1998, vol. 1, pp. 98–99).

Korean thought is not limited to or interchangeable with the worship of Tan'gun (the mythical founder of Korea) or Taejonggyo (大倧敎),[30] but is a complex collection of ideas that include Confucianism, Buddhism, and Taoism. While Confucianism, Buddhism, and Taoism were imported as foreign ideas, they have been absorbed, digested, and expanded by Korean scholars over a thousand years, and there is no reason not to present Korean scholarship on these philosophies as part of Korean thought. Therefore, Yi emphasizes the need to research and publish on these subjects.

Yi categorizes the history of Korean thought into three periods: the Kyŏnghak (經學, Classical Studies) and Sajanghak (詞章學, Literary Studies) period that spans the Three Kingdoms Period to the late Koryŏ dynasty, the Lihak or Tohak (理學 or 道學, Neo-Confucian) period that spans the late Koryŏ to the Qing invasion of Chosŏn in 1636 or Pyŏngja horan (丙子胡亂), and the Silhak (Practical Learning) period that ranges from the post-Qing invasion period to the late Chosŏn dynasty. The first period was an early stage where the Three Kingdoms imported Confucianism and established the foundation for Korean studies (國學), and applied the study of Confucian canons and classics to Korea's politics, systems, and customs. The second period, the Neo-Confucian stage, saw the influx of Buddhism and Taoism, which initiated a wave of self-reflection on the philosophical principles and ultimate goal of action. The last period was preceded by two invasions from Japan and Manchuria, and this crisis led Korean Confucian scholars to undergo a reflective self-examination of the nature, objective, and methodology of Confucianism, which resulted in the rise of Practical Learning (Yi 1998, vol. 1, pp. 115–22).

Before delving into his research on Korean thought, Yi points out the errors of previous scholarship on Chosŏn's Neo-Confucianism and reexamines Hyŏn Sangyun's (玄相允, 1893-?) "theory on the merits and vices of Confucianism" (儒敎功罪論).[31] According to Yi, the eight vices of Confucianism identified by Hyŏn—apart from "the decline in industrial capability"—are not problems inherent to Confucianism but problems that arise in the process of applying Confucianism to everyday life. In other words, these vices result



from the failure to properly master and employ the spirit of Confucianism (Yi 1998, vol. 1, pp. 11–42).

Contrary to the dominant academic opinion that viewed Neo-Confucianism as doctrinaire and as the cause of Chosŏn's downfall, Yi asserts that "the fundamental spirit of Neo-Confucianism is devoted to promoting ethical action in everyday life" (Yi 1998, vol. 1, p. 421). To illustrate, in examining the Confucian debates over Wuji (無極, Infinite Principle) and Taiji (太極 Great Principle), Yi points out that Chinese Confucians Zhu Xi (朱熹) and Lu Jiu-yuan (陸九淵) clashed over their different interpretations of the ontology of the universe, while Yi Ŏnchŏk (李彥迪) and Cho Hanpo (曹漢輔) disagreed over the definition and practice of Dao (道). He names Yi Ŏnchŏk as the first Korean Confucian scholar to incorporate both the theory and practice of Neo-Confucianism, Tohak (道學) in particular, into his philosophy (Yi 1998, vol. 1, pp. 162–203).

According to Yi Sang-ŭn, Yi Ŏnchŏk's spirit of Tohak was later inherited by Yi Hwang and Yi I (李珥), the two great masters of Korean Neo-Confucianism. Yi Hwang's *Ten Diagrams to Become a Sage* inherits the spirit of 'reaching the heavenly pattern-principle up above' (上達天理), while Yi I's the *Essentials of the Study of Sages* (聖學輯要) inherits the spirit of 'learning from the lives of humans down below' (下學人事). Yi Sang-ŭn, as part of his research on Yi Hwang that includes *The Life and Work of T'oegye Yi Hwang (T'oegye ŭi Saengae wa Hangmun)*, traces the theoretical evolution of the spirit of Tohak. While his research on Yi I was left unfinished, it is presumed that the research focused on the practical application of Tohak.

Yi Sang-ŭn observes how Yi Hwang and Yi I's spirit of Tohak was later carried on by the T'oegye school and Yulgok school of Confucian scholars, and how this evolved into Silhak in the late Chosŏn period. Neo-Confucianism rejects the notions of 'falsity' (虛) and 'emptiness' (空) and instead pursues the actual practice of moral principles—namely, the 'reality/practicality' (實) that is in the very title of Silhak (實學). According to Yi Sang-ŭn, scholars in the late Chosŏn period later inherited and improved this spirit of Silhak by advocating the importance of external practice (事爲) as well as the internal cultivation of the mind.

More specifically, Yi focuses on specific scholars—Chŏng Tojŏn (鄭道傳), Yi Ŏnchŏk, Yu Hyŏngwŏn (柳馨遠), and Chŏng Yakyong (丁若鏞)—to analyze the socio-historical context and ideological background of Silhak. He states that these scholars all pursued the realization of the true laws of nature (天理), and the only difference was in their proposed methods of actualizing this true law. In other words, the emergence of late-Chosŏn Silhak is the result of different scholars focusing on different aspects of Neo-Confucianism, and not the result of a radical change in the aim and nature of the philosophy (Yi 1998, vol. 1, pp. 367–88).

Hence, Yi strove towards an objective and rational evaluation of Neo-Confucianism and Silhak by examining their fundamental spirit, approach to life, and way of study and discipline through an analysis of philosophical literature and documents. According to Yi, Silhak and Neo-Confucianism did not contradict or oppose each other, but ran parallel and supplemented each other. He maintains that "Neo-Confucianism without Silhak is an empty shell, Silhak without Neo-Confucianism is blind and unconditional" and that "Neo-Confucianism is the ideology that leads Silhak toward the right direction, and Silhak is the methodology that provides the specifics of Neo-Confucianism" (Yi 1998, vol. 1, p. 429).

Prior to Yi's research, it was conventional to deem Neo-Confucianism as doctrinaire and Silhak as incompatible with the Zhu Xi school of thought. However, Yi bases his arguments on historical evidence to challenge these prevalent misconceptions that fracture the genealogy of Korea's intellectual history, be it Japan's colonialist account of history that views Neo-Confucianism as the cause of political strife or the Yushin (the Revitalizing Reform) regime's discourse on national autonomy,[32] Yi considered them to be inaccurate accounts that impeded the proper understanding of intellectual history and delayed the genesis of a new academic climate.

After designating Silhak's dedication to moral action as the main spirit of Korean Confucianism, Yi worked towards a comprehensive understanding of the history of Korean intellectual thought by focusing on how this spirit carried on and evolved across time. Yi finds the significance of his project to be in the fact that he gleaned historical and moral lessons from the lives of his predecessors. Silhak's spirit of morality is a valuable intellectual asset, and Yi argues that inheriting and elevating this spirit can help address the problems of modern society and create a 'third path' that guides civilization towards coexistence and collective prosperity.

## 5. Conclusions

The unjust reality of Korea under Japanese colonial rule was the starting point of Pak and Yi's philosophical investigations. Pak studied Heidegger with hopes of interrogating the problems of reality and finding a detailed solution, but suspended his study after realizing that Heidegger's philosophy is limited to individual awareness and lacks specific and active engagement with reality. Yi was drawn to Hu Shi's movement to 'systematize national heritage' and dreamed of initiating Korea's own reevaluation of traditional thought from a modern perspective, but was impeded by Korea's colonial reality.

After Korea's liberation, Pak concentrated on establishing a system for the study of Western philosophy, logic in particular, and directed his attention to the dialectical logic of Hegel. He analyzed modern Western philosophy through the Hegelian framework of thesis, antithesis, and synthesis, and asserts that Eastern philosophy is the synthesis of existentialism and the philosophy of science. On the other hand, Yi defends the need to reinterpret Eastern classics through a rational method that combines scientific analysis and empathetic reading, and applies this method to a thorough examination of the evolution of Chinese Confucianism while focusing on inheriting and supplementing the tradition.

In the mid-1950s, while Korea's academia came to reflect on their previous dismissal of Korean philosophy, Pak and Yi were able to confirm the status and duty of Eastern philosophy during their visit to the U.S. and shifted their research to the study of traditional thought. Pak's research confirms that elements and issues addressed by Western philosophy already existed in traditional philosophy, and affirms the creative spirit of Korean philosophy that pursues the dialectical synthesis of ethics and economy. He strove to uncover a sense of cultural pride by presenting Korean thought as one of the alternatives to modern Western philosophy. Meanwhile, Yi analyzed and examined the spirit and methods of Neo-Confucianism and Silhak to demonstrate that Silhak inherited the essence of Neo-Confucianism's pursuit of moral action, and urged for revisions to be made in conventional academic discussions of Silhak.

Pak pursued the birth of a new, modern Korean philosophy, and concentrated his research on the creative spirit of traditional thought and the dialectical incorporation of foreign ideas based on a strong sense of national autonomy. The National Charter of Education was part of Pak's specific contributions towards contemporary Korean thought, as it was the process of bringing forth a philosophy that is alive and responsive to reality. Meanwhile, Yi attempted to address the problems of modern society by inheriting and elevating Silhak's spirit of moral action. This naturally extended to his public disapproval of the rigged presidential election of 1960 and his sharp criticism of the misconceptions prevalent in Korea's academia.

Thus, this article surveyed the academic trajectories of Pak and Yi—trajectories that has both divergences and overlaps, in the context of New Korean Confucianism. Pak borrowed Hegel's dialectics to reinterpret traditional thought as the synthesis of an introspective thesis and extroverted antithesis, and attempted at genesis of Korean philosophy that merged both traditional and foreign thought. On the other hand, Yi understood the necessity to reinterpret and reevaluate traditional Eastern philosophy from a new, modern perspective; established traditional Confucianism as a logical and objective field of study; and demonstrated the potential of Korean Confucianism to be applicable to modern society.[33] From this perspective, their philosophical investigations into the reinterpretation

and modern application of traditional Confucianism constitute a uniquely Korean model of New Confucianism, namely New Korean Confucianism.

**Author Contributions:** Writing—original draft preparation, X.-A.G.; writing—review and editing, S.-Y.C. All authors have read and agreed to the published version of the manuscript.

**Funding:** This research was funded by the 2022 Korean Studies Grant Program of the Academy of Korean Studies (AKS-2022-R-037), the Chey Institute for Advanced Studies' International Scholar Exchange Fellowship for the academic year of 2020–2021, and 2022 National Social Sciences Funding for the Western Project "A Critical Study of Zhu Xi's Theory in Modern Korea (1876–1945)." (22XZX014) [2022 年国家社科基金西部项目 " 近代朝鲜对朱子学的批判研究(1876–1945)" (22XZX014) 的阶段性成果].

**Institutional Review Board Statement:** Not applicable.

**Informed Consent Statement:** Not applicable.

**Data Availability Statement:** Not applicable.

**Conflicts of Interest:** The authors declare no conflict of interest.

## Notes

[1] 'New Confucianism' is alternatively called as the 'Contemporary New Confucianism (當代新儒學)' or 'Modern New Confucianism (現代新儒學)' but here I adhere to the more neutral term, 'New Confucianism,' denoting what follows after Neo-Confucianism. There may be some doubts whether New Confucianism is a religion or not; it will eventually lead to the discussion whether Neo-Confucianism, or even Confucianism, is a religion of not, which is beyond the discussion of this article. Still, from the 'thick' perspective or definition of world religion, Confucianism may be understood as an all-encompassing way of thinking and living—a worldview, a social ethic, a political ideology, a scholarly tradition, and a way of life—that inextricably entails human-centred religiousness.

[2] The definition of '(Contemporary/Modern) New Confucianism' is still debated among scholars of Eastern philosophy. Fang Keli (方克立) presents a very broad definition that not only includes all attempts to reaffirm the values of Confucianism under modern conditions and restore the prominence of the Confucian tradition, but also includes attempts to merge Confucianism with Western learning as a way to envision a practical future for China's culture and society (Jing 1993). Meanwhile, Wei Zheng-tong (韋政通) identifies seven characteristics of New Confucianism that include acknowledging Confucianism as the foundation and tradition of Chinese culture, regarding the history and culture of China to be a single spiritual entity, emphasizing the role of respect and empathy in understanding history, sharing consciousness of crisis regarding culture, and having religious overtones. (Wei 1990, p. 215). Similarly, Cheng Chung-ying (成中英) suggests six conditions necessary to New Confucianism that include understanding the origin and philosophical system of Confucianism; emphasizing the wisdom, spirit and practice of Confucianism; and suggesting ways to both interpret and solve modern problems regarding knowledge, will, and behavior. (Cheng 1996, p. 264). After 1949, the hub of New Confucianism moved to Hong Kong and Taiwan. The field even advanced in areas such as the United States, and continued to expand its scope of research. There are many ways to categorize the evolution of New Confucianism, but this paper will subscribe to the categorization proposed by Zheng Jiadong (郑家栋), and consider the first generation as Liang Shuming (梁漱溟), Zhang Junmai (張君勱), and Xiong Shili (熊十力); the second generation as Feng Youlan (冯友兰), He Lin (賀麟) and Qian Mu (钱穆); the third generation as Mou Zongsan (牟宗三), Tang Junyi (唐君毅), and Xu Fuguan (徐復觀); and the fourth generation as Du Weiming (杜維明), Liu Shuxian (劉述先), and Cai Renhou (蔡仁厚) (Zheng 1993, pp. 18–29). Recently, a significant volume of research related to Chinese New Confucianism has been conducted, especially as Chinese Confucianism became more familiar in Western academia (Bell 2010; Fan 2011; Sole-Farras 2013).

[3] Western philosophy made its way into Korea in the late Chosŏn period, when Catholicism was first introduced to the peninsula. It became well-circulated around Korea by the 20th century, as can be seen in *Kaebyŏk* (開闢), a periodical from the 1920s that published multiple articles introducing the philosophies of Nietzsche, Kant, Rousseau, Russell, and Hegel. Nevertheless, this paper will define the above as the first generation of Western philosophy researchers because: (1) these scholars specialized in Western philosophy by studying primary sources and texts, (2) this was when the first department of philosophy that provided professional training was founded in Korea, and (3) these scholars were the first to engage in a systematic process of writing and publishing works on philosophy. (Park 2015, p. 201).

[4] Both Pak and Yi are mostly known for their contribution to the National Charter of Education (國民教育憲章) and the declaration against the rigged presidential election of 1960. Since the mid-1990s when people started reconsidering the acceptance of Western philosophy and the establishment and development of Korean philosophy, researchers have paid attention to their philosophical thoughts. The *Pak Chonghong jŏnjip (Complete Works of Pak Chonghong)* was published in 1998, by Minŭmsa, and in the same year, the *Yi Sang-ŭn sŏnsaeng jŏnjip (Complete Works of Master Yi Sang-ŭn)* was published by Yemunsŏwŏn. Subsequently, with the support from the Korean Yŏl-am Memorial Foundation and the Korean Society of Confucian Studies, further studies on both Pak's and Yi's philosophical thoughts were conducted. As a result, scholarly books such as *Hyŏnsilkwach'angjo (Reality and Cre-*

*ation)* Vol. I (1998) and Vol. II (2001), *Pak Chonghong ch'ŏrhagŭi chaejomyŏng (Reassessing the Philosophy of Pak Chonghong)* (2003), *Yi Sang-ŭn Sŏnsaenggwa Han'guk Shinyuhak (Yi Sang-ŭn and Korean New Confucianism)* (2006), *Han'guk Hyŏndae Tongyangch'ŏrhakkwa Yi Sang-ŭn (Korean Contemporary Eastern Philosophy and Yi Sang-ŭn)* (2017) were published. Based on these studies, a significant number of attempts to make their overall philosophical thoughts more logical and systematical again emerged.

[5]  This article will not go over the details about how they diverged over political issues later in their careers, especially related to former president Park Chung Hee's (1917–1979) regime. Some scholars criticize Pak being silent over, if not actively contributing to Park's tyrannical dictatorship, while Yi actually wrote a protest letter to Park that he should keep the promise to democritize the political system. This article will focus on their trajectories converging at creating the foundation for the Korean Philosophy.

[6]  There are only a handful of studies on Pak and Yi's philosophy in the context of New Confucianism. For Yi, see Chung (2005); Choi (2010); Hong (2016); and Gao (2021). For Pak, see Gao (2018, 2020).

[7]  The term 'nation'(民族, *min-jok*, literally people-kinship) appears in the writings of Korean students studying abroad in Japan, circa 1910, as a synonym for blood relatives. By expanding the sense of kinship beyond blood relatives and creating a notion of nationship, they aspired to Korea's independence. (Park 2010, pp. 78–84)

[8]  Now Seoul National University. As Japan's sixth imperial university, Keijō Imperial University was established by the Japanese Government-General to supplement the colonial education system. Preparatory courses were founded in 1924, and regular courses were founded in 1926. After Korea's liberation in 1945, the university briefly changed its name to Kyŏngsŏng University before announcing the plan to establish Seoul National University in September 1946.

[9]  After graduating with a degree in Philosophy from Tokyo Imperial University, Miyamoto Wakichi was dispatched to Germany and England in 1923 as an overseas researcher for the Ministry of Education (文部省). After returning to Japan in 1925, he contributed to the compilation of the Iwanami Dictionary of Philosophy and Thought (岩波哲學思想事典), and later joined Keijō Imperial University as a professor in April 1927 where he taught courses including Introduction to Philosophy, History of Western Philosophy, Epistemology, and German Idealism.

[10]  Pak writes: "Surrounded by the introspective solitude, I see no other option than to find my way back to society. However, how can a new society unfold? Compared to the ample criticism of the decadence of everyday secular reality, I cannot help but notice the lack of action towards constructing a new society shaped by fundamental existence. I cannot find in Heidegger the course to specific action." (Pak 1998, vol. 2, p. 293).

[11]  Apart from the publications of Ilban nollihak [General logic] in 1948 and Inshik nollihak [Epistemic logic] in 1953, Pak did not get the chance to publish his project during his lifetime. Pyŏnjŭngbŏp chŏk nolli [Dialectic logic] was published posthumously in 1977 by his students.

[12]  Pak wrote, "Essence is the primary negation of being, and notion is the secondary negation—namely, the negation of the negation of being. Therefore, notion is the reconstructed being. It is essence returned to being in a form of immediacy. However, by going through the process of sublation, being and essence do not exist in their initial form. Thus notion is primary to being and essence, or immediacy and reflection, and in this manner being and essence are the two causes of notion. Consequently, being and essence go under or collapse into notion, and are preserved. Notion is the product of being and essence, and therefore these two are preserved inside notion in an elevated state. In this regard, notion is the synthesis of being and essence; it is their truth" (Pak 1998, vol. 3, p. 520).

[13]  Pak's dichotomy of introspection and extroversion bears a strong resemblance to Kōji Tanaka's (田中幸治) reading of Dostoevsky's Crime and Punishment. Both of them draw on concepts such as introspection and extroversion, centrifugal and centripetal force, to support the same argument, which indicates Tanaka's influence on Pak. (Cho 1982, pp. 19–20).

[14]  Yi published "Sunja ŭi inshim doshimnon" [Xunzi's theory of the human spirit and the spirit of Dao] in 1954, "Koja ŭi innaeŭioe sŏr e kwanhan haesŏk" [An analysis of the debate on the internality and externality of benevolence and righteousness in "Gaozi"] in 1955, "Maengja ŭi sŏngsŏnsŏr e taehan yŏn'gu" [A study on Mengzi's theory of human nature] in 1955, "Kongja haksŏr ŭi chungshim gaenyŏm ŭl nonham" [A treatise on the main concepts of Confucian theory] in 1963, "Chungyong dohae" [A diagram of the Middle Way] in 1969, and "Sŏngnihak ŭi sŏngnip kwajŏng" [The establishment of Neo-Confucianism]. He also published papers that dealt with much more generic issues of Chinese philosophy, including "Tongyang munhwa rŭl chaeronham" [A re-examination of Eastern culture] in 1956, "Chesa ŭi ŭiŭi" [The significance of ancestral rites] in 1956, "Yugyonŭn chonggyo in'ga" [Is Confucianism a religion] in 1960, "Sanghagwan gwa ch'abyŏl gwan" [Hierarchy and discrimination] in 1962, "Yuhak ŭi ponjil gwa shidae jŏkŭng" [The essence of Confucianism and its adjustment to new times] in 1972, and "Hyumŏnijŭm esŏ pon tongyang sasang" [Eastern philosophy as seen from the perspective of Humanism] in 1976. These papers are all collected in the third volume of Yi Sang-ŭn sŏnsaeng chŏnjip, Chinese Philosophy.

[15]  From 1960 to 1966, these lecture notes were published in volume 3 to 8 of *Han'guk sasang* under the title "Chungguk ch'ŏrhaksa kangŭi" [Lectures on the history of Chinese philosophy].

[16]  Yi writes, "This is because we must add new interpretations based on modern insights, but never distort the original argument by practicing contextomy, and engage in scientific arguments based on sufficient bibliographic evidence." (Yi 1998, vol. 4, p. 63).

[17]  Here, the 'scientific' method refers to non-subjective, non-mystical means to obtain the core or principle: If Neo-Confucian scholars, especially during the Ming–Qing dynasty, interpreted and understood classics through a rather personal realization, insights, or sudden awakening, the scientific and objective method relied on earlier commentaries (closer to the original, less

[18]    influenced by Buddhist and foreign influences), a more coherent interpretation based on newly discovered manuscripts, etc. It is closer to the 'evidential learning' of later Qing dynasty Confucianism than to Song–Ming Neo-Confucianism.

[18]    This bears strong resemblance to the New Confucian theme in "Wei Zhongguo wenhua jinggao shijie renshi xuanyan" (爲中國文化敬告世界人士宣言, A Manifesto on the Reappraisal of Chinese Culture). In fact, Yi maintained academic ties with the leaders of the very same manifesto, including Tang Junyi (唐君毅), Xu Fuguan (徐復觀), and Mou Zongsan (牟宗三).

[19]    For example, in his analysis of the *Analects*, Yi points to the frequency of righteousness (義) mentioned in the text to demonstrate how it is unreasonable to view righteousness as the central theme of the *Analects*. He suggests scholars focus on the context and definition of righteousness, instead. (Yi 1998, vol. 3, pp. 629–54).

[20]    This is in line with the philosophical methodology that Feng Youlan set forth in his writings on *Xin lixue* (新理學 *New Rational Philosophy*). Feng integrates the 'positive (正) method', a logical mode of analysis that deals with the contents and objects of metaphysics, with the 'negative (負) method,' a mode based on intuitionism (直覺主義) towards the ineffable mystic realm.

[21]    Feng Youlan argues that the history of Chinese philosophy does not fit into the Western categories of ancient, medieval, and early modern philosophy, and suggests the categorization into *Zixue* and *Jingxue* as an alternative. Yi adopts both modes of categorization.

[22]    Hu Shi uses the term "reaction" (反動) in his *Outline of the History of Chinese Philosophy* to chart the origins, developments, and influences of ancient schools of thought. Hu Shi surmises that Laozi's politics of non-action (無為) was a reaction against the central regime's intervention, and Mozi's emphasis on practical application was a reaction against Confucianism's fixation on oughtness over practicality.

[23]    This eerily mirrors the career of Hu Shi and his unfinished manuscript on the history of philosophy. Like Yi, Hu Shi only published the first volume of his project on ancient philosophy.

[24]    Pak stayed at University of Minnesota from September 1955 to August 1956, and Yi stayed at Yale University from August 1956 to June 1957.

[25]    This early stage of research that focuses on the similarities between Eastern and Western philosophy continued until the early 70s (Huh 1996, p. 224).

[26]    For more on Pak's research on Korean Buddhist thought, see Gao (2019, pp. 115–38).

[27]    This core idea was sprouting in the early formation of Tonghak, which was later actualized in Ch'ŏndogyo (天道敎). In fact, this specific phrase was proposed by Son Son Pyŏnghŭi in 1905 after Tonghak was actually transformed into Ch'ŏndogyo.

[28]    Pak provides an example of the Chosŏn Confucain scholar of Practical Learning Yi Wŏn'gu (李元龜 or 一叟, 1758–1828), who depicted the ethical and economical as an inseparable whole, and likened it to a conjugal relationship (Pak 1998, vol. 5, pp. 256–73).

[29]    Here, the sense of national autonomy has a similar overtone to Juche (autonomous) thought which became the state ideology of North Korea, since Juche developed in response to the intrusion of foreign ideas and the need to produce philosophy that makes sense in the peculiar and unique situation in the Korean peninsula. Discussion on how Pak and Yi's philosophy emphasizing national autonomy exactly differs from North Korean Juche Ideaology is beyond the limitation of this article. Still, it is worth noting that Pak's emphasis on autonomy is more on the level/depth of Korean traditional philosophy having the sufficient power and potential to open itself to the world and interchange ideas at the same level, rather than exporting a unique Korean-style social policy or a certain government ideology.

[30]    Tan'gun is the mythical founder of Korean nation and Taejonggyo is the Korean new religion founded cir. 1904–9 based on Tan'gun myth that he forms the triology with Heaven (Hanul) and the father, Hwanung, and manifested himself in Korea in order to save the ethnic Korean people from oppression and suffering.

[31]    According to Hyŏn's *Chosŏn yuhaksa* [The history of Confucianism in Chosŏn], the merits of Confucianism include: (1) encouraging the pursuit of becoming a noble man (君子); (2) the veneration of ethics and morals; and (3) respect towards integrity and loyalty. The vices include: (1) blind imitation of sinocentrism (慕華); (2) empty political strife; (3) the harms of familism; (4) hierarchical classism; (5) feebleness that arises from overindulgence in literature (文弱); (6) the decline in industrial capability; (7) blind submission to authoritative orders; and (8) antiquarianism (復古) (Hyŏn 2003, pp. 22–29).

[32]    Yi focused on criticizing Japan's imperialist accounts of Korean history during the 1950s, but shifted to criticizing the discourse on nation propagated by the Yushin dictatorship. Strongly influenced by Hu Shi, Yi viewed these as distorted views on Korean history that had to be revised based on thorough research and investigation (Kim 2014, p. 74).

[33]    As briefly mentioned in the Chapter 4, Pak saw the potential of Korean tradition to contribute to the world philosophy in their rich discussions on the topic of human dignity, existentialism, and philosophy of science. Yi also brought out the original interpretations of Korean Confucian scholars in their commentaries on the Four Books and Five Classics of China. Still, their lifetime devotion to protecting the Korean tradition and stressing its capacity to be a generalizable philosophy could be seen as lacking theoretical innovations, or specific content that brings Eastern and Western thoughts together. Some may point out that their study was philology, not philosophy. Nonetheless, their awakening of their identity as Korean, not merely a part of greater Chinese cultural sphere, drove them to systematize the national heritage of Korea, just as their Chinese and Japanese counterparts did to their own tradition. The very act of unearthing and re-arranging the old wisdoms of Korea so that later Korean scholars

could participate in academic discussions with world philosophers and religious thinkers is in itself a pioneering work, which is the core characteristic of New Confucianism.

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
