# Peer review of "Characterizing ‘New Korean Confucianism’: Focusing on Pak Chonghong and Yi Sang-ŭn’s Life and Thought"

_religions, doi:10.3390/rel14020138_

Round 1
Reviewer 1 Report
Interesting but I would have like to have seen at least one specific example from the philosophies of both Pak and Yi which show how they brought Eastern and Western thought together. Also, this article hints at but doesn’t mention explicitly the differences in the ways Pak and Yi thought about Park Chung Hee.
I found a few typing errors:
p. 1 line 25 and its victory over
29 took up should be to take up.
32 absorption of the Western cultural
p. 3 line 79 both of their should be both in their
footnote 5 There are only should be There is only
p. 4 line 127. Pak hence suspended his
p. 6 footnote 13 Yugyon ŭn chonggyo in'ga should be Yugyonŭn
p. 7 footnote 18 He suggests scholars to focus delete to
p. 8 line 293 promoted how the works
should be promote the idea that the works
p. 10 line 350 leads Pak to the examine delete to
p. 10 footnote 28 philosophy emphasizing the national autonomy exactly differs
it is worth to note should be it is worth noting
p. 11 footnote 29 Tan’gun is the mythical founder of Koran Peninsula Should be Korean nation.
line 436. Taoism is normally spelled Taoism these days.
Also on p. 12. line 449 Taoism should be spelled Taoism.
p. 12 line 451 two invasions from Japan and Korea Korea should be replaced by Manchuria.
The same term romanized as Dohak in one 443 is romanized elsewhere on this page correctly as Tohak.
p. 14 line 550 National Chart of Education should be National Charter of education
line 560 attempted at a genesis of Korean philosophy delete a
line 564 Korean Confucianism to be applicable

Author Response
Thank you for all the very helpful comments and suggestions.
1) one specific example from the philosophies of both Pak and Yi which show how they brought Eastern and Western thought together --> I don't think their New Korean Confucianism had reached at such a level that could effectively merged the two traditions together. They just have opened the way of New Korean Confucianism that still is ongoing process. I have added this remark on the last footnote 33.
2) Mention explicitly the differences in the ways Pak and Yi thought about Park Chung Hee. -> I have added on the footnote 5, but didn't go too explicit about it. It is worth dealing in another article.
3) All your helpful comments on the typing and grammatical erros have been corrected. Thank you (as in the file)
Reviewer 2 Report
The paper is very interesting and also significant because it critically presents New Confucianism in Korea, which is not very well known in Western scholarship, although at least in sinology and among experts in Chinese philosophy, the current of Chinese New Confucianism is already very well researched. In this regard, I believe that the author has not considered most of the recent sources dealing with this current. In this respect, she or he can improve her/his introduction in which she/he presents the ideational background of Modern New Confucianism. Some more specific remarks can be found below:
1. On the threshold of the 20th century, China's cultural pride was hurt not only by Japan, but even more by the European invaders who forced China into a semi-colonial position.
2. The author writes: "Pak, for instance, studied philosophy at Keijō Imperial University Japan, while Yi studied abroad at Peking University, where he was exposed to various Western theories and the complexities of Chinese academia such as New Confucianism." I wonder if for a Korean scholar like Pak, studying in Japan really did not count as studying abroad? Of course, Korea had been colonized by Japan since 1910, but Japan was probably still considered a foreign power? And was it really cheaper to study in Japan than in China, as the author implies on page 4?
3. Although the essay is a very good introduction to Korean New Confucianism, I wonder how it might relate to the field of religious studies? New Confucianism has nothing to do with religion, while this journal, according to its title, is about religious studies. Does this paper really fit into this content area? The author should explain the connection.
4. The overall methodology and structure of the article are good, but it is still "just" a review and does not bring any theoretical innovations. These should be elaborated in the conclusion, which as it stands is not really a conclusion, but only a summary.
If the author could correct or change these minor details, the article would be much better and I could easily recommend it for publication.
Author Response
Thank you very much for your helpful suggestions and comments.
0) the author has not considered most of the recent sources dealing with this current. -> I have added some recent books on New Contemporary Confucianism in footnote 2, although this is far from being sufficient. The definition of New Confucianism in the footnote 2 may seem outdated, but this is the most prevalent and well-known definition in Korean academics today. We are still in process of updating the recent discussions.
1) even more by the European invaders -> I moderately changed the Introduction part. Page 1 (28-30)
2) It was a typo-mistake!!!. Thank you for pointing that out. Keijo Imperial University in Korea, not Japan. It became Seoul National University. I added the changes in the footnote 8.
3) This article is part of the special issue entitled, "Korean Confucianism" that covers not only topics in the studies of religion, but also philosophy, ethics, political ideology. - I added the comments in the footenote 1.
4) "just" a review and does not bring any theoretical innovations. -> I have added this part in the footnote 33. I don't believe neither Pak nor Yi had reached at the summit of bringing out 'theoretical innovation' that merge East and West together. They only paved the way, laid the foundation for the 'Korean' Confucianism.
Thank you.